



# Turning regret into future disaster preparedness with no-regrets

Joy Ommer[1,2], Milan Kalas[2], Jess Neumann[1], Sophie Blackburn[1], Hannah L. Cloke[1,3]

[1]Department of Geography and Environmental Science, University of Reading, Reading, RG6 6UR, United Kingdom
[2]KAJO s.r.o., Bytca, 01401, Slovakia
[3]Department of Meteorology, University of Reading, Reading, RG6 6UR, United Kingdom

*Correspondence to*: Joy Ommer (j.ommer@pgr.reading.ac.uk)

**Abstract.** Global efforts are focusing on long-term preparedness for disasters highlighting the need for taking well-informed decisions in advance to avoid panic behaviour when a disaster strikes. Taking well-informed decisions includes the evaluation of the potential outcomes of a decision or action to avoid regretting them afterwards. Yet, little is known about what we regret
about our actions and inactions in the context of disasters. Using the responses of a survey disseminated in flood affected areas in German in 2021, this study dives into the regrets of citizens and the reasons for their regrets. The results showed the that participants only regretted preparedness actions when they threatened their life, but foremost, participants regretted their inaction. Overall, the results indicate the need for promoting long-term preparedness which can be supported with no-regrets actions which in addition need to be easy-to-implement. Furthermore, the need for integrating actions supporting psychological
preparedness was identified. To increase citizens preparedness motivation, their self-responsibility needs to be enhanced which could be achieved through fostering collective action.

## 1 Introduction

*'I was woken up by the rising water as I swam across the room on my couch.'*

*(Original: 'Ich wurde von dem steigenden Wasser geweckt, als ich mit meiner Schlafcouch durchs Zimmer schwamm.')*


After a flood, we would probably reflect on the moment when we woke up on the couch because it was floating through the room. In this moment, we would start thinking what if I had received a warning and prepared for the flood. The reflections on the past and thinking about the 'what if' can make us regret decisions and actions that had negative outcomes, or actions we failed to take (Feldman and Chen, 2019; Feldman et al., 1999; Gilovich and Medvec, 1994; Zeelenberg et al., 2002). Regret is
an emotion of blaming ourselves, but regret and the experience itself, also supports us in adapting for future flooding (Hung, 2020; Kuang and Liao, 2020; Zeelenberg and Pieters, 2007).

The floods in Germany of July 2021 have left many regrets but also starting points for enhancing future disaster preparedness. Although, the event was forecasted well in advance at both European and national level (A. H. Thieken et al., 2023), the floods took hundreds of citizens by surprise because they did not receive any warning or did not take warnings seriously (Fekete and
Sandholz, 2021). Those citizens who did not expect flooding, did not have had time to prepare and, therefore, were overwhelmed by the water entering their homes (Lemnitzer et al., 2021; A. H. Thieken et al., 2023). The lack of preparedness



together with people taking risky actions such as driving through flood water or going downstairs into flooded basements caused a high number of lost lives (A. H. Thieken et al., 2022).

The floods further reminded us that many citizens have a rather reactive or flood defensive mindset, rather than a proactive
one (Ommer, Blackburn, et al., 2024; Surminski & Thieken, 2017). Rare events like this disaster are deeply uncertain and therefore, need to be adapted to in advance rather than taking action only after a warning (which sometimes may not arrive) (Marchau et al., 2019). A proactive mindset can enhance our ability to act fast when we receive a warning which otherwise perhaps results in irrational, reflexive, or panic decision taking (L. Geaves et al., 2023; Xenidis & Kaltsidi, 2022). Hence, for taking good decisions on preparedness actions, we need time to evaluate the potential impact of our actions to ensure that we
will not regret them in the future (Robinson and Botzen, 2018; Sunderrajan and Albarracín, 2021; UNDRR, 2022).

This raises the research question of which actions do we actually regret and why do we regret them. Commonly communicated preparedness actions include preparing the home for intruding water by moving valuable things upstairs, installing pumps in a basement, or preparing an emergency kit (Kreibich et al., 2011; Martins et al., 2019). These actions can be performed in a relatively short time, for instance, after receiving a warning. Although, these actions taken are very valuable for protecting our
home and properties, they were recently claimed as 'weak preparations' where we are 'blindly following' instructions (Katsikopoulos, 2021). In fact, (proactive) disaster preparedness shall target the strengthening of knowledge and capabilities 'to effectively anticipate, respond to and recover from the impacts' (Sendai Framework Terminology on Disaster Risk Reduction, 2023) which may not be always achieved with so-called weak preparedness actions. Hence, stronger and long-term preparedness actions rather include developing a (household) emergency plan and practising it (Katsikopoulos, 2021).

Considering that the uncertainty about whether we will be affected by a hazard or not (especially, if we have no prior hazard experience) is one major barrier to long-term preparedness, an interdisciplinary strategy for decision-making under uncertainty could be applied: the no-regrets approach (Marchau et al., 2019). First integrated into climate policies in the 1990s, the no-regrets approach fosters taking actions which are robust in different future scenarios (i.e., no, low, or high impact hazards). According to Heltberg et al., (2009), actions shall firstly, not be regretted in any future scenario, and secondly, not be costly,
and entail benefits. These values have motivational factors to take the actions since our decisions are driven by the aim to avoid regret but also by economics and benefits which are representing a reward (Sunderrajan and Albarracín, 2021).

The no-regrets decision-making strategy was later adopted in disaster risk reduction research (Plume, 1995; Heltberg et al., 2009; Debele et al., 2023), but has not been applied in context with individual disaster preparedness. Gaining a better understanding of the regrets linked to flood preparedness can help shape advice on preparedness behaviour for citizens. For
this purpose, this study explores the flooding event in Germany in 2021 from a regret perspective. Using an online survey disseminated in flood affected areas, this study dives into citizens' preparedness actions before the event and for the future. The main objective is to gain insights into what participants regretted, or not, and why. Acknowledging that regret is a cognitive process and therefore, highly subjective, this study aims to derive a broad overview on potential regrets of citizens. Secondly, the outcomes of the survey will then be used to form recommendations for long-term disaster preparedness and the suitability
of the no-regrets approach as a framework for individual disaster preparedness.





To learn more about the flooding event, Section 1.1 provides a summary about the floods and impacts. Section 2 presents the survey design and data analysis. The results providing insights into the regrets and no regrets are discussed in Section 3 and a conclusion towards long-term disaster preparedness is provided in Section 4.

## 1.1 The floods in Germany in 2021

The low-pressure system 'Storm Bernd' brought heavy precipitation in Western Europe between 12th and 15th July 2021 which cascaded into flooding and caused devastating impacts (Kreienkamp et al., 2021; Lemnitzer et al., 2021).

Germany (and its neighbouring countries) experienced severe rainfall after a three-week-period of wet days (Dietze et al., 2022). While heavy rainfall hit many parts of the country, the federal states Rhineland-Palatinate and North Rhine-Westphalia experienced particularly high amounts of precipitation causing local flooding. The two states are located in the western region

of Germany bordering to the Netherlands, Luxembourg, Belgium, and France. In these states, many small and medium sized rivers exceeded their banks during the flooding event (Lehmkuhl et al., 2022).

Heavy precipitation of up to 180 mm within 72 hours led to various types of flooding (Dietze et al., 2022; Junghänel et al., 2021; Lehmkuhl et al., 2022). The initial high saturation level of soils led quickly to surface runoff and pluvial flooding (Dietze et al., 2022). While the runoff on hillslopes transformed into small streams forming gullies (Lemnitzer et al., 2021). Flash

floods occurred in the middle hills' catchments where steep slopes are a common landscape feature (A. H. Thieken et al., 2023). Additionally, water reservoirs filled up quickly and proved danger to their dams and the downstream population (Lehmkuhl et al., 2022). Lastly, urban fluvial flooding occurred in cities along rivers and streams (A. H. Thieken et al., 2023). In Germany, about 162 km$^2$ were inundated, primarily affecting the agricultural sector with 88 km$^2$ of flooded agricultural land (He et al., 2022). Overall, the floods led to devastating damages of EUR 32 billion (Mohr et al., 2022). The damage to roads,

bridges and other critical infrastructure further complicated evacuation and emergency response (Fekete and Sandholz, 2021; Koks et al., 2022). Most importantly, more than 180 people lost their lives and hundreds of people were injured or displaced (Dietze et al., 2022; A. H. Thieken et al., 2023). According to an evaluation in the federal state North Rhine-Westphalia (A. H. Thieken et al., 2022), most people lost their lives in their cars, on the street, in a basement, or on the ground floor. Most of these people drowned in the flood waters, a few lost their life due to heart failure, and two because of burn injuries from oil-

fired heating.

The event was referred to as 400-year event but highlighting the fact that these kinds of events can occur more often (Kreienkamp et al., 2021). The results further suggested an influence of climate change on the intensity of the rainfall event and future ones. According to another study, land cover changes, for instance in America, could further intensify these rainfalls in the future (Insua-Costa et al., 2022).



## 2 Methods

### 2.1 Online survey

To gain a better understanding on the perspective of affected citizens on this event, an online survey was designed to gather a spatially wide collection of responses. To give these citizens a voice, the survey (Supplementary Material) encompassed primarily open questions regarding the flood source, risk estimation, preparedness, response, early warning, issues that were perceived and suggested solutions for these, and basic demographic questions (age, living situation, and postcode). The survey was designed in two languages (German and English). After approval by the SAGES Ethics Committee of the University of Reading (February 2022), the survey was open from March to July 2022 and invited flood affected citizens (18 years and older) from the two states Rhineland Palatinate and North Rhine-Westphalia to share their experiences. It was disseminated using Microsoft Forms via social media channels such as Facebook, Twitter, LinkedIn, and WhatsApp. The nature of the design of the study and dissemination strategy could lead to biases (i.e., Ong et al. (2023)) in age groups, risk awareness, or the personality of participants as it may have promoted the participation of generally more active and engaging people.

### 2.2 Data analysis

The responses were stored in Microsoft Excel and pre-processed which included the translation of responses in English, the correction of postcodes, and the adding of municipality and district names (based on the postcodes).

Descriptive statistics were used to analyse basic questions regarding age structure and living situation, location, and flood experience. In total, 438 responses were collected. The majority of participants (87,7%) were living in North Rhine-Westphalia at the time of the flooding and 12,3% in Rhineland Palatinate. 65% of the participants were aged between 25 and 54 years but also covered the age groups 18-24 years (6%), 55-64 years (19%), and 65 years and older (9%). The age structure of the survey participants is comparable to the German national age structure of 2022 (Population in Germany, 2024), but shows a slight overrepresentation of the age group 25 to 54 years. Almost two thirds of the participants were owning a house in July 2021, and about one-fifth were living in a rental apartment. Other participants were living in a rented house (7%), owning a flat (4%), or living with their parents or guardians (3%).

Open questions were analysed qualitatively utilising thematic analysis (Braun and Clarke, 2006). This allowed a deeper insight into the survey results by identifying overarching themes. Applying the thematic analysis, all questions were analysed to distil themes that appear across these questions. The workflow included the familiarisation with the responses followed by an initial coding in Nvivo (release 1.7.1) which highlighted the themes of inaction and regret. Using Microsoft Excel, these themes were explored in more depth by manually coding the responses into i.e., reasons for inaction. The results of the thematic analysis are presented and discussed in Section 3 and concluded in Section 4.



## 3 Results

Overall, participants implied different regrets about their preparedness behaviour, especially, about the actions they did not take – their inaction. Regret was expressed in what they would do differently if another flood approaches in the future. This showed that the participants have been evaluating their actions and inactions and probably thought about what they could have done (differently). This reflection and the question of 'what if' is a typical process that can lead to regret (Feldman et al., 1999; Zeelenberg et al., 2002). What participants regretted and why is discussed in the following Section 3.1. In contrast, Section

3.2 discusses what participants did not regret and indications for why.

### 3.1 What do we regret?

Overall, participants mentioned that they undertook a variety of short-term emergency measures (Kreibich et al., 2011; Martins et al., 2019) such as preparing the house and basement for potential water intrusion, moving valuable furniture, documents, photos, and more upstairs, preparing emergency escape bag packs, storing food, and filling water canisters. Interestingly, none

of the participants mentioned that they regretted having taken any of these actions. Only if the action caused a threat to the life of the participant, then regret was expressed.

*'I cleaned out the basement.*
*Which, in retrospect, was very dangerous. I wouldn't do that anymore.'*
*(Original: 'Ich habe den Keller ausgeräumt.*

*Was im Nachhinein sehr gefährlich war. Das würde ich nicht mehr machen.')*

This can be explained by the regret theory that when an action had or might have had a negative outcome, we start thinking about the 'what if' – imagining what we could have done differently to achieve a more positive outcome (Zeelenberg et al., 2002). In the above quote, the participant recognised afterwards that going into a flooded basement can be very dangerous,

indeed drowning in the basement was one of the major threats of this flooding event (A. H. Thieken et al., 2022). Causes of death were also linked to driving or walking in flood water which was regretted by another participant.

*'Stay at home and stop trying to drive the car.'*
*(Original: 'Zuhause bleiben und nicht mehr versuchen mit dem Auto zu fahren.')*


Similarly, negative outcomes were associated with the trust in the early warning. Flooding was largely unexpected by the survey participants primarily because of untimely, late, or no warning. Roughly half of the participants did not receive any warning considerably in advance of the event (24h or more). Asking the participants when they received the first warning, about 20% did not receive any warning before the water arrived, 26% noted that they received warning only a few hours before

and about 14% were 'warned' by the arriving or entering water itself. Not receiving warning in time left several participants



in the situation that they had to evacuate immediately. In this context, participants highlighted that they regret to trust in the dissemination of early warning and the support by the local authorities which they see as the cause for this stressful and dangerous situation they found themselves in.

160         *'No warning' – 'Escape' – 'Trust no one' – 'Do everything (differently)'*
*(Original: 'Keine warnung' – 'Flucht' – 'Keinem vertrauen' – 'Alles (anders machen)')*

*'I no longer rely on warnings! In case any come!!!*
*Keep an eye on the surroundings/nature myself.'*
165         *(Original: 'Ich verlasse mich nicht mehr auf Warnung[en]! Falls welche kommen!!!*
*Selber die Umgebung/Naturi m Auge behalten.')*

Their intent for the future was to not trust nor depend on local authorities and warnings. Trust is an important pillar for the relations between citizens and authorities which is especially needed in emergency situations (Earle, 2010). However, trust is

very dynamic and fragile (Seebauer and Babcicky, 2018); thus, it is often observed that after a devastating event, the trust in authorities diminishes if expectations are not reached (Seebauer and Babcicky, 2018; Whitmarsh, 2008). This case shows how the regret has initiated that participants are willing to take more responsibilities which is a typical effect of regret (Zeelenberg et al., 2002). At the same time, these participants now aim to be more proactive by taking measures in advance and being more attentive to the nature and environment to detect changes to avoid surprises in the future.


*'Now, one is aware of what 200L/sqm means. With similar amounts, I would have packed my suitcase long ago and would*
*move to higher elevations for safety.'*
*(Original: 'Jetzt ist man sich bewusst was 200L/qm bedeutet. Bei ähnlichen Mengen hätte ich schon längst den Koffer*
*gepackt und würde mich in Höhere Lagen in Sicherheit bringen.')*

In this context, one participant implied regrets about the person's own knowledge and threat appraisal. In this case, the person perhaps did not have enough knowledge or information about what 200ml of rain would mean and hence, had to take decisions under greater uncertainty (Marchau et al., 2019).

185         *'Public warnings are not reliable.' – 'No trust in those in charge' – 'Escape' – 'Take all*
*measures in advance yourself to avoid being caught off guard.'*
*(Original: 'Auf die öffentlichen Warnungen ist kein Verlass.' – 'Kein Vertrauen in die Verantwortlichen.' – 'Flucht' –*
*'Selbst im Vorfeld alle Maßnahmen treffen um nicht überraschend zu werden.')*



In disaster situations, it is important to take well-informed decisions which involves considering the potential outcomes of a decision (Sunderrajan and Albarracín, 2021; UNDRR, 2022). These surprise moments have left participants no or minimal time; thus, decisions were made in panic or reflexive leading to irrational action taking (L. Geaves et al., 2023; Xenidis & Kaltsidi, 2022).

*'I tried to save things and forgot important things. One acts irrationally.'*
*(Original: 'Ich habe versucht Dinge zu retten und habe wichtige Dinge vergessen.*
*Man handelt irrational.')*

Overall, many regrets were related to time as participants would have like to prepare earlier ('*früher*'), in time ('*rechtzeitig*'),
or immediately ('*sofort*' '*direkt handeln*' '*zeitig*').

*'In the short time, less than an hour before the event, there was nothing more one could do.'*
*(Original: 'In der kurzen Zeit, keine Stunde vor dem Ereigniss, konnte man nichts mehr machen.')*

The majority of regrets expressed by participants were related to inaction. Missing the chance to take actions because of different reasons is another common cause for regret (Gilovich and Medvec, 1994). 29.6% of the participants wrote that they did nothing ('nichts', 'nix') as preparedness. This preparedness inaction was primarily linked to the lack of time to take action because of the unexpectedness of the event. More reasons for not preparing were the fact that people were e.g., sleeping (since the flooding started during the night in some areas); they were not at home; they did not understand the warning properly; they
could simply not imagine an event like this; they did not know what to do; or they could not act because they were not the legal owners.

Inaction regrets were found in regard to flood mitigation and preparedness measures, evacuation, seeking information, and helping others. Considering the debate in research about whether actions or inactions are more regretful (Feldman and Chen, 2019), it can be concluded that in this real-life experience, inactions were regretted more as participants perhaps wanted to
take actions but could not because there was not enough time.

Inaction was further explained by the fact that participants did not take the received warnings seriously which they regretted afterwards.

*'Unfortunately, I did not pay enough attention to the warning, so [I prepared] nothing.'*
*(Original: 'Ich habe der Warnung leider nicht genug Beachtung geschenkt, also nichts [vorbereitet].')*

One reason for not taking the warning seriously was that there have been too many warnings, especially, also considering the recent Covid-19 pandemic for which the same warn app was utilised. This effect is commonly referred to as alert fatigue



(Potter et al., 2018; Roberts et al., 2022). Another participant who did not take warning seriously mentioned that when the

flood arrived, the person tried to react but gave up at some point. Experiencing the flood and the regret has triggered that the

person would take warnings more seriously and take different actions next time. This learning from floods is very common

and acknowledged in research (Carone et al., 2019; Köhler et al., 2023; Kuang & Liao, 2020; Kuhlicke, Masson, et al., 2020).

## 3.2 What don't we regret?

Despite the many regrets that are summarised above, a few participants clearly stated that they did not regret anything about

their actions.

As surprise and stress situations can cause reflexive behaviour, it is important to be psychologically prepared to stay calm

(APS, 2018). One participant mentioned that they were worried but managed to stay calm despite the fact that they would have

liked to evacuate if they had received warning.

*'We were somewhat worried but kept calm. Had we known beforehand*

*that it would be much worse than predicted, we would have left our home.'*

*(Original: 'Wir waren schon etwas besorgt, aber haben Ruhe bewahrt. Hätten wir vorher gewusst, dass es viel schlimmer*

*wird als vorhergesagt, hätten wir unser Heim verlassen.')*

Another participant highlighted that acting very prudently was not to be regretted.

*'Actually nothing, I proceeded very prudently, however,*

*prior information from the municipality would have been helpful.'*

*(Original: 'Tatsächlich nichts, ich bin sehr besonnen vorgegangen, allerdings wäre*

*eine vorherige Information seitens der Gemeinde helfend gewesen.')*

This quote also reflects an earlier finding that people did not claim any regrets on the actions they took unless they (almost)

had negative outcomes. Reversing this finding, it can be assumed that all actions that were taken somehow improved the

overall outcome or at least did not have any negative effects. Furthermore, taking actions can avoid the regret about failing to

do something. It can be related to a statement from another study 'at least doing something' (Nalau et al., 2021) meaning that

doing something is better than doing nothing. The preparedness actions taken by participants were not regretted; thus, they can

be categorised as no regret actions which are better than inaction.

No regrets were expressed by participants who stated that they have done everything they could do within their (perceived)

abilities to act and self-organise (Kievik and Gutteling, 2011).


*'Tried to stop it, saved the most important things, then saved myself.'*



*(Original: 'Versucht es aufzuhalten, das Wichtigste gerettet, dann selbst gerettet.')*

*'Packed things, brought family, neighbours, and friends to safety.'*
*(Original: 'Sachen gepackt, Familie, Nachbarn und Freunde in Sicherheit gebracht.')*

Helping others to prepare, evacuate, or similar was one major aspect which was not regretted by participants. Helping others was also one action that was taken by many participants, in general. Interestingly, even if these actions may have caused a threat to the person's own life, they were not regretted. This helping behaviour and not regretting it may be explained by different psychological backgrounds such as anticipating the guilt of not having helped someone in need, because it may bring us pleasure to help others, or because we have a moral responsibility to help others (Erlandsson et al., 2016).

*'But I was also standing up to my chest in water to get people out. I would do that again.'*
*(Original: 'Aber ich stand auch bis zur Brust im Wasser um Leute da raus zu holen. Das würde ich wieder tun.')*

Contrarily, some participants believed that they could not have done anything to prepare for an event like the one in July 2021, and that it is only possible to respond reactively.

*'That's the force of nature, one can only react.'*
*(Original: 'Das ist Naturgewalt, man kann nur reagieren.')*

In contrast to the above quote, intended behaviour changes and taken measures imply that the flooding experience evoked a more proactive mindset. Increasing risk awareness and learning from flooding is a common process building on the reflections of past events (Kuang and Liao, 2020; Kuhlicke et al., 2020). In this regard, it was mentioned that it is important to have a plan for actions to be taken in emergency cases which can be an easy step towards preparedness.

*'Create an emergency plan and then execute it.'*
*(Original: 'Einen Emergency Plan erstellen und dann durchführen.')*

Having an emergency plan and drills are considered stronger preparedness actions and, simultaneously, present proactive actions that can be implemented any time even without an imminent hazard (Katsikopoulos, 2021; Marchau et al., 2019).



## 4 Conclusion

This study explored the flooding event in Germany in 2021 from the perspective of regret to gain a deeper understanding on citizens (no) regrets on disaster preparedness actions to derive lessons learnt towards long-term preparedness actions. In this

study we analysed what was regretted (or not) and identified reasons that led to the regret. The results of this study suggest the following implications for disaster preparedness.

Firstly, short-term emergency measures are valuable but long-term preparedness is more important. Participants used various emergency measures to prepare their home in a hurry. These actions were referred to as weak actions (Katsikopoulos, 2021), but not useless as none of the participants regretted having taken those. However, more awareness needs to be raised on the

fact that taking these measures in the very last-minute such as preparing the basement when flood water is already intruding, can pose a threat to life and therefore, could become regrettable. In the sense of 'at least doing something' (Nalau et al., 2021), taking actions was not regretted as people perceived that they did (everything) what they could. In contrast, citizens regretted their inaction because they sort of failed to do things they could have done, but there was not enough time. Overall, the results highlighted the need to take the following actions in advance – basically from today onwards: developing an emergency plan

including evacuation scenarios, learning to understand the environment better to be able to spot changes or to know what forecasted values will be like in reality.

Secondly, actions need to create a feeling of awareness, responsibility, and independence. Citizens were greatly dependent on authorities in this flooding event (Ommer et al., 2024). This dependency and their trust in authorities to manage the flooding caused great regrets when the expectations were not met, and this created a difficult situation for the participants. This is a

common issue also in other countries i.e., in the UK (Cologna et al., 2017; Thorne, 2014). To anticipate these impasses citizens were in, and to avoid increasing distrust in authorities, actions should support the creation of awareness on risks, build environmental knowledge, and loosen dependencies of citizens while increasing their feeling of responsibility.

Thirdly, long-term preparedness needs to integrate actions that increase the psychological preparedness of citizens. A few participants showed that staying calm in this kind of stress situation where the water is imminent, it is very important to take

decisions and not just act reflexive or in panic (APS, 2018).

Fourthly, actions need to be within the capabilities of citizens. The results showed that citizens have taken actions that were within their abilities and in some cases had to give up. Therefore, actions need to be easy-to-implement.

Lastly, many actions that were taken in advance of the flooding were focusing on helping others in various situations and people did not regret this even if their own life was at risk. This finding acknowledges the importance of supporting family,

friends, neighbours, and even unfamiliar people. In response to this finding, individual long-term preparedness could be enhanced by focusing on collective action.

Summarising the above findings, it was highlighted that citizens need to be motivated to take long-term preparedness actions in order to cope with future (unexpected) hazards and their impacts. The findings of this study suggest that the no-regrets approach could be a suitable framework combining emergency preparedness and, foremost, long-term preparedness due to the

robustness of actions in different scenarios, the no regret factor in case no hazard may be happening, and its motivational elements. However, in addition to the introduced characteristics and to ensure actions are taken and not regretted, the findings of this study showed that no-regrets actions must be easy-to-implement; thus, citizen are able to take them and support the idea of collective action as a motivational and enhancing factor for individual preparedness and self-responsibility, respectively. Overall, this study has highlighted that regret and the experience of a flood can increase future preparedness which is

responding to a large amount of findings from other studies. Yet, the question is whether we really need to experience and regret flooding first before starting to consider long-term preparedness? The no-regrets approach can pave the way towards long-term preparedness of citizens, but more research is needed on how to facilitate citizens walking this way.

**Data availability**

The participants of this study did not give written consent for their data to be shared publicly, so due to the sensitive nature of

the research, the survey data is not available.

**Author contributions**

Joy Ommer (conceptualisation, investigation, visualisation, writing, review & editing); Milan Kalas (conceptualisation, supervision, review & editing); Jess Neumann (conceptualisation, supervision, review & editing); Sophie Blackburn (conceptualisation, supervision); Hannah L. Cloke (conceptualisation, supervision, review & editing).

**Competing interests**

The authors declare that they have no conflict of interest.

**Acknowledgement**

Hannah L. Cloke acknowledges funding from the UKRI Natural Environment Research Council (NERC) The Evolution of Global Flood Risk (EVOFLOOD) project Grant NE/S015590/1.

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
