# Peer review of "Turning regret into future disaster preparedness with no-regrets"

_EGUsphere, 2024_

## Author Response (AR1)

**Response to reviewers**

Reviewer 1:

We thank you for your thoughtful and constructive review of our manuscript! We note your suggestions and will provide more details on the knowledge gap as well as on the methodology. We aim to integrate your recommendations for the results section as far as the survey data can provide this information. Furthermore, we will improve the discussion around short and long-term preparedness, but it will be challenging to add the suggested visualisation due to the nature of the data. Also, it will be challenging to clearly distinct between personal choices and the institutional context for the same reason. However, we will aim to clarify the pre-conditions. We will restructure the results section and improve the conclusion in line with your recommendations. Once again, we thank you for your valuable feedback and for helping us strengthen the manuscript.

| Comment | Response |
|---|---|
| Context of the Study: The introduction provides a clear and comprehensive overview of the topic of regret in the context of disaster risk management. However, the study would benefit from an additional paragraph in the introduction that reflects on the knowledge gap this study addresses and its context within existing literature. | We added a paragraph to strengthen the research gap in section 1. |
| Methodological Details: The survey developed by the authors contains very interesting and useful questions and demonstrates a thorough approach to collecting relevant information on subjective regret experiences. It would be beneficial to include more details about the survey in the main text, such as a table summarizing the key questions considered in this study. Additionally, visualizing some of the collected contextual information (e.g., questions 2, 3, and 7 from the survey) and referring to this in the results section would add clarity. Clarification is also needed on whether any filtering of the responses was necessary or if all 438 responses were suitable for consideration. | The main questions were added (section 2.2) and some insights into filtering was added. |
| Result Details: The anecdotal insights from the survey provide interesting reflections on regret by citizens but are somewhat general. A more in-depth analysis, such as examining correlations between experienced impacts, feelings of preparedness, and regret, could add depth to the analysis. While the authors mention a relationship between regret and (near-)failure of actions, it would be beneficial to discuss this more explicitly. | We added more information to most of the quotes on suggested topics. |
| The authors refer to two groups of actions, including long-term preparedness. In the introduction, different types of preparedness | We extended the discussion on the weak and strong actions (section 4.3) but as anticipated, we could not add any discussion on the |

| | |
|---|---|
| measures (e.g., placement of furniture, pumps, emergency kits) are defined as 'weak' preparations, while (proactive) disaster preparedness (mentioned as long-term preparedness) includes emergency plans and drills. The paper would benefit from a more detailed discussion on the limits and benefits of each group in light of the given event, especially considering its low occurrence probability. Discussing the nuances in the measures taken and the associated regret for each type would add valuable context. For instance, a 2D visualization (x: no regret, regret; y: no damage – fully destroyed house) could illustrate which measures were generally perceived with more or less regret depending on the experienced damage. Furthermore, discussing specific types of measures and their associated regret would inform the recommendations regarding future preparedness more effectively. | nuances in the measures due to the lack of data. |
| Finally, it could be an idea to make a distinction between (in)action of citizens because of personal choice and the role of the institutional context. The authors discuss the importance of emergency plans and the insufficiency of the warning timing/comprehensiveness which lies clearly outside of the capabilities/responsibilities of an individual citizen but play a crucial role to create the pre-conditions for no-regret preparedness by the citizens. | Unfortunately, there is not much data available for this suggestion. The only distinction could be the lack of warning (institutional context) and not taking warnings seriously (personal choice) but as the latter group is so small it would not make sense to discuss this in depth. |
| Result Section Structure: The distinction between actions that were regretted and those that were not is useful. However, the authors should adhere more strictly to this separation. For example, the initial sentences under "What do we regret" actually discuss measures that were not regretted. Additionally, the sections could be streamlined to align with the recommendations the authors intend to provide, such as actions that are regret-free, and the role of awareness and information access in (in)action. The discussion of various reasons for inaction, including insufficient access to or understanding of information, should be consolidated for coherence. | We reshuffled the results sections to make it more coherent. The reasons for inaction were extended with some numbers but as this was a non-mandatory question, not all participants replied to the specific question. |
| The conclusion section needs significant revision. It is uncommon to use the conclusion for anything other than reflecting on the key findings from the study and discussing limitations or remaining knowledge gaps. The current draft mixes summary and new information and does not at all reflect on | The conclusion was reorganised and split into discussion and conclusion. In this process, the other suggestions were addressed. |

| | |
|---|---|
| remaining knowledge gaps or weaknesses of the present study design/data-set. A clear distinction should be made between the analysis of survey data and the authors' reflections on answering the research question ("recommendations for long-term disaster preparedness and the suitability of the no-regrets approach"). Additionally, some conclusions drawn seem unsupported by the analysis provided. The four recommendations for no-regret future preparedness require further detail. The authors should differentiate actions based on their primary purpose within the DRM Cycle and their feasibility for citizens. Reflecting in more detail on the survey's learnings to support a nuanced set of recommendations would be very beneficial. | |

Reviewer 2:

We appreciate the time and effort you dedicated to evaluating our work and providing detailed feedback! Your insights are valuable to us and will help enhance the clarity and impact of our study. We thank you for the general comment on linking the paper more closely to risk governance in Germany which we focused on in another paper (Risk social contracts: Exploring responsibilities through the lens of citizens affected by flooding in Germany in 2021) and will add information on flood warnings. Furthermore, we will split the conclusion section as suggested by you and address your specific comments. We thank you again for your feedback!

| Comment | Response |
|---|---|
| The paper could benefit from further context about responsibilities of the citizens compared to the government, and the creation of a separate discussion section with the paper's implications for disaster preparedness. Section 1 sets out the research context well, but the paper could benefit from additional framing around legal responsibilities of flood risk and response in Germany. Whilst there was a lot of detail about the hazard and exposure of the 2021 floods, there was no detail about the flood risk governance structure in Germany – specifically the balance of the responsibilities of preparedness between citizens and the government. Outlining this would allow the reader to understand that the German Federal State requires that residents hold legal responsibility for protecting their property, and therefore understand the implications if | We added a paragraph (to section 1) reflecting the outcomes of this survey in the governance & trust context: Risk social contracts: Exploring responsibilities through the lens of citizens affected by flooding in Germany in 2021 https://doi.org/10.1016/j.pdisas.2024.100315 |

| | |
|---|---|
| residents are failing to prepare/protect sufficiently for flooding. Using this framing would also allow the authors to link their results of dependence and trust between citizens and local authorities back to this context in Section 3. | |
| In addition, there is an absence of a discussion section, whereby the implications of this study on regret for long-term disaster preparedness should have been discussed. Splitting up the content of Section 4 into a discussion (recommendations) section and a conclusion would provide more clarity on how this paper contributes to wider literature and to German flood governance. More detail is also needed on the five implications for disaster preparedness. This would be enhanced through suggestions for how the recommendations could be implemented by either citizens or the government; such as explicitly suggesting how long-term preparedness could be enhanced by collective action, and what needs to change about the current system of preparedness for this to happen. | We revised the conclusion (section 5) and added a discussion (section 4) discussing the insights from the study and resulting implications. |
| The paper would benefit from adding some nuance into the context in Section 1 around the lack of preparedness being due to the lack of flood warning. Whilst flood warnings often do lead to an increased chance of preparedness, some citizens may not have the social capital or mobility to act upon these warnings. One or two sentences outlining this challenge would add to the paper. | Thank you for the suggestion, we added some information on this to Section 1. |
| On line 92-95; please specify how land cover changes in America could intensify further rainfalls in the future (from the Insua-Costa et al, 2022 reference), as that was slightly confusing with no further information. | It is a very insightful study, we recommend reading. However, we removed the sentence since extending it would shift away from the focus of the paper. |
| The methods section could explicitly outline some of the most significant survey questions in the main text to make it more accessible to the reader, rather than just listing them the supplementary materials. These could be embedded alongside the quotes, to make it clear what the citizen was answering about. | The main questions were added (section 2.2). However, we did not add the questions to the quotes because it would be either too much more information or if only using e.g. Q5, readers must always go back and look up the question. |

---

## Author Response (AR2)

**Comment(s):**

Dear authors,

Thank you for submitting your manuscript to NHESS. Editing the manuscript in response to the comments made by reviewers 1 and 2 has improved its quality considerably. Based on your responses to the reviewers and the changes made to the manuscript, I have decided that the manuscript can be considered for publication in NHESS after minor revisions. Before the manuscript can get published, please make sure that you also include the limitations of your study in the discussion (e.g. limitations of the sampling strategy, external validity of the results etc.). The manuscript would also benefit from a careful check for spelling and grammar including re-phrasing awkward sentences such as the first sentence: 'After a flood, we would probably reflect on the moment when we woke up on the couch because it was floating through the room.'

Best wishes,
Viktor Roezer

**Response:**

Dear Mr. Roezer,

We thank you very much for your suggestions! We revised sentenced which might be of awkward structure and added a section on limitations (4.4) to the discussion as requested.

Please, let us know if you need anything else from us!

All the best,

Joy Ommer